# COMBINE AND COMPARE:
# GRAPH RATIONALE LEARNING WITH CONDITIONAL NON-RATIONALE SAMPLING

## ABSTRACT

Traditional Graph Neural Networks (GNNs) assume an ideal distribution of independent and identically distributed (i.i.d) data, a rarely met condition in real-world datasets. Therefore, how to address distribution shifts between training and testing sets becomes paramount in GNNs. Recently, the rationale learning method has garnered much attention as a graph generalization method. It first divides the graph into label-related rationale subgraphs and label-unrelated non-rationale ones. Then, it creates diverse training distributions by combining different non-rationales with rationales. Finally, by exploring the invariant rationales across training distributions, the performance of GNNs facing out-of-distribution (OOD) graphs is boosted. However, this method still faces two problems: (*i*) when combining non-rationales with rationales, it commonly *randomly* samples a non-rationale and combines it with the rationale. This may inadvertently produce duplicate samples. (*ii*) the relationship between the rationales, non-rationales and labels is not properly considered, where non-rationales and labels should be de-correlated compared to the rationales. To address these problems, we propose a Combine and Compare (CoCo) with non-rationales for Graph Rationale Learning method with the conditional non-rationale sampling. Specifically, from the framework of rationale learning, CoCo first employs the diverse sampling method to sample non-rationales, avoiding sampling duplicate non-rationales. Besides, we introduce a non-rationale progressive hard sampling method to de-correlate hard non-rationales and labels, enhancing the model's discrimination ability. Extensive experiments on both benchmarks and synthetic datasets demonstrate the effectiveness of our method for OOD graphs. Code is released at https://anonymous.4open.science/r/CoCo-5410/.

## 1 INTRODUCTION

Graph neural networks (GNNs) (Li et al., 2022; Xu et al., 2019; Scarselli et al., 2008) have merged as a fundamental model to solve realistic problems in different fields such as social networks (Fan et al., 2019; Barabâsi et al., 2002) and biological networks (Xinyi & Chen, 2018; Eisenberg & Levanon, 2003). By leveraging the inherent graph structure, GNNs have demonstrated remarkable efficacy in capturing the intricate relationships and dependencies in these real-world scenarios.

Despite the enormous success, incumbent works have made spectacular achievements based on the assumption that samples across both train and test sets obey the independent and identically distributed distribution (i.i.d). However, this distribution is overly idealistic for datasets collected from the real world. When confronted with out-of-distribution (OOD) graphs, GNNs' performance significantly deteriorates, thereby constraining their application in real-world scenarios (Ying et al., 2019). Considering Figure 1, we make predictions regarding the motif type by leveraging the graph consisting of motifs and bases subgraphs. In the training set, there are two types of frequently occurring graphs (i.e. *Circle*-motif with *Ladder*-base and *House*-motif with *Tree*-base). This prevailing *Motif-Base* combinations may mislead traditional GNNs (Figure 1(a)) to learn the statistical dependency between motifs and bases for excellent performance, instead of exploring the real relationship between graphs and labels. As a result, when dealing with *Circle-Tree* or *House-Ladder* (i.e., the OOD data), the probability of predicting it to be circle or house decreases.

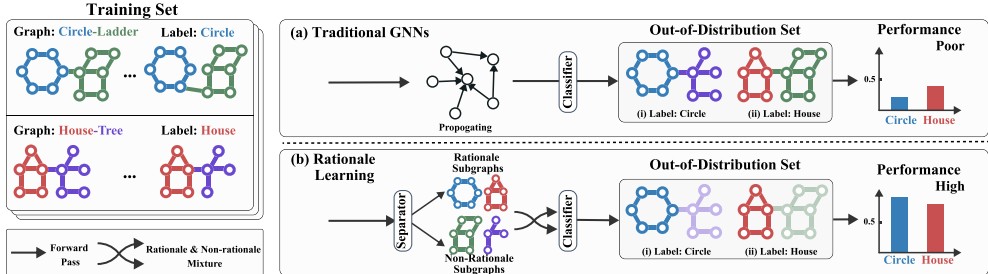

Figure 1: An illustration of graph OOD problem and generalization. (a) Traditional GNNs heavily rely on spurious correlations in the dataset to make predictions, such as the statistical dependency (frequently occurring *Circle-Ladder* graphs), leading them to prone to mistakes when faced with OOD data (*Circle-Tree*). (b) The rationale learning framework consisting of the Separator, Rationale & Non-Rationale Mixture and Classifier modules. Since this method identifies invariant rationales across diverse training distributions, it can alleviate the graph OOD problem.

To solve this problem, numerous methods have been proposed. Among them, the rationale learning method (Chang et al., 2020; Wu et al., 2022b; Miao et al., 2022; Liu et al., 2022) (Figure 1(b)) has received increasing attention, which extracts a subset of the graph as the rationale to best support model prediction while keeping the rationale invariant across different data distributions. As shown in Figure 1, the circle subgraphs in the *Circle-Ladder* graph are referred to the rationale.

Figure 1(b) presents the framework of rationale learning for graph generalization that involves three modules: (1) **Separator**, which can be formulated as function $f_S(g_i)$, extracting rationale subgraphs $r_i$ (label-related features) from the input graph $g_i$ accompanied by the rest non-rationale subgraphs $e_i$ (label-unrelated features). (2) **Rationale & Non-Rationale Mixture**, which can be formulated as function $\mathrm{AGG}(r_i, e_j)$, combining rationale subgraphs $r_i$ with other non-rationales $e_j$ $(i \neq j)$ to create multiple training distributions to increase the diversity of data. Meanwhile, since the rationales do not change, labels of the mixed data remain unchanged. (3) **Classifier**, which can be formulated as function $f_C(\cdot)$, predicting the graph class based on both the rationale generated by Separator and the mixed data ($f_C(\mathrm{AGG}(r_i, e_j))$). Finally, by approaching the real rationales that are invariant across different distributions, this rationale learning approach can improve the generalizability of GNNs. Although this approach is promising, it still suffers from the following two problems:

Firstly, In the Rationale & Non-Rationale Mixture, most approaches (Fan et al., 2022; Sui et al., 2022; Liu et al., 2022) employ the non-rationale based augmentation methods to create diverse training data, which "*randomly*" sample a non-rationale in a batch to combine with the rationale. Although this approach enables the rapid creation of training distributions, there is a risk that such randomized operations may sample similar non-rationales to the separated non-rationales. For example, assuming *Ladder* is the separated non-rationale of the *Circle-Ladder* graph, after the randomized operations, the combined non-rationale may be still *Ladder*. In such a setup, too many "duplicate" samples may be created, which have the potential to reduce the effectiveness of the model training process and further negatively affect the final results.

Besides, in the Classifier, it is common to employ both the rationale and mixed data for prediction, focusing on the invariance of the relationship between the rationale and label across various train distributions. However, this type of approach ignores the partial order relationship between rationales and non-rationales, where non-rationales and labels should be de-correlated compared to the rationales. Therefore, this method may result in the separated non-rationales still containing part of the information of rationales, thereby diminishing the effectiveness of the rationale learning.

To address the above problems, in this paper, following the framework presented in Figure 1(b), we propose the **Co**mbine and **Co**mpare (**CoCo**) with non-rationales for Graph Rationale Learning method with conditional non-rationale sampling. Specifically, we first generate rationale and non-rationale subgraphs using a Separator. Then, in the Rationale & Non-Rationale Mixture module, we introduce a diverse sampling method for non-rationale based augmentation. Different from the "*randomized*" operations, diverse sampling is to select non-rationales that significantly differ from the anchor non-rationales to combine with rationales, avoiding sampling "duplicate" non-rationales. Next, in the Classifier, to de-correlate non-rationales and labels, we develop a non-rationale progressive hard sampling for exploiting the partial order relationship of rationales and non-rationales. It employs a percentile-based strategy to gradually screen out a set of hard non-rationales similar to

the anchor rationales, enhancing the model's ability to discriminate between the two and ensuring a stable rationale learning process. Finally, we conduct extensive experiments on both benchmarks (Hu et al., 2020) and synthetic datasets to validate the effectiveness of CoCo on OOD graphs.

## 2 RELATED WORKS

**Graph generalization.** A foundational assumption in Graph Neural Networks (GNNs) is that training and testing set are independent and identically distributed distribution. Regrettably, this assumption seldom aligns with the intricate realities of real-world scenarios, resulting in a sharp performance degradation on OOD data. Faced with this challenge, recent research efforts have focused on the generalization capabilities of GNNs (Garg et al., 2020; Knyazev et al., 2019). Some research addressed the OOD at the node-level classification, such as EERM (Wu et al., 2022a; Fan et al., 2022). This paper focuses on the graph-level generalization (Miao et al., 2022). Recently, researchers utilized rationalization techniques to identify rationales subset of the input graph for graph classification by invariant leaning (Li et al., 2022; Wu et al., 2022b) and graph augmentation (Liu et al., 2022). DIR (Wu et al., 2022b) focuses on causal rationales that remain invariant through controlled random interventions on the training distribution. Liu et al. (2022) augments original graph by *random* removing and replacing non-rationales to strengthen the rationale representation learning against the noise signals brought by the non-rationale subgraphs. Altough the effectiveness of rationale learning in enhancing generalization, the *random* interventions on distribution or graph data augmentation are obviously thoughtless. In the light, our work rethinks the graph rationale learning's objective and proposes a reasonable sampling method for the objective accordingly.

**Data Sampling**. One key component of our framework is to sample non-rationales in comparisons with rationales and for non-rationale based augmentation, which is most relevant to sampling strategy technology applied in some domains like natural language processing (Mikolov et al., 2013), recommendation (Rendle & Freudenthaler, 2014), contrastive learning (Robinson et al., 2020) etc. One classical approach is static sampling strategies which sample instances based on a predefined distribution, such as uniform and popularity distribution corresponding to random sampling (Rendle et al., 2009) and popularity-based sampling (Caselles-Dupré et al., 2018; Mikolov et al., 2013) respectively. However, static methods cannot adjust to model training, suffering from low quality of samples. Adaptive sampling (Rendle & Freudenthaler, 2014) was proposed later, such as DNS (Zhang et al., 2013) which dynamically selects hard samples that are difficult for the current model to discriminate. Most work (Robinson et al., 2020; Ge et al., 2023) uses sampling methods to select samples based on the objectives of different tasks and model training. Nonetheless, few works have explored this issue in the study of rationale & non-rationale. In this work, we design an adaptive sampling strategy for non-rationale variables, considering both the rationale & non-rationale partial order learning and non-rationale based augmentation.

## 3 METHODOLOGY

In this study, we employ the graph classification task to evaluate the effectiveness of our method in addressing the OOD problem at the graph level. We first show the problem definition (Section 3.1). Subsequently, following the rationale learning framework (Section 3.2), we provide a comprehensive description of our conditional non-rationale sampling method (Section 3.3). Finally, we present the optimization and inference procedures employed in our study (Section 3.4).

### 3.1 PROBLEM DEFINITION

*Graph Classification with Rationalization.* A graph classification task involves assigning a category or label to a given graph based on its structural properties or attributes. Consider a set of labeled graphs $\mathcal{G} = \{(g_i, y_i)\}_i^n$, where $g_i = (\mathcal{V}, \mathcal{E})$ represents the $i$-th graph, $\mathcal{V}$ is the set of nodes, $\mathcal{E}$ is the set of edges and $y_i$ is its corresponding label. The goal of graph classification with rationalization is first to learn a separator $f_S(g_i)$ to generate the probability of each node being rationale $M_i \in \mathbb{R}^{|\mathcal{V}|}$. Then, given the graph node representation $H_i$ which is encoded by any GNN, we can further obtain the rationale subgraph representation as $r_i = \text{Pooling}(M_i \odot H_i)$. Finally, we employ a classifier $f_C(r_i)$ to yield the task results based solely on $r_i$. For example, given a motif type graph consisting of *Circle* and *Ladder* in Figure 1, the goal is to extract the *Circle* subgraph in the latent space to make predictions.

*Graph Generalization.* Given the graph training set of $n$ instances $\mathcal{G}_1 = \{(g_i, y_i)\}_i^n$ from training distribution $\mathcal{P}_1 = (g, y)$ and the testing set $\mathcal{G}_2$ from testing distribution $\mathcal{P}_2 = (g, y)$, where $\mathcal{P}_1 \neq \mathcal{P}_2$. Note that the testing distribution is unknown during the training stage. The goal is to train a separator $f_S(\cdot)$ and a classifier $f_C(\cdot)$ on training set $\mathcal{G}_1$ that achieve generalization on testing set $\mathcal{G}_2$.

$$f_S^*, f_C^* = \arg \min_{f_S, f_C} \mathbb{E}_{g, y \sim \mathcal{P}_2} \left[ \ell \left( f_C(f_S(g)), y \right) \right]. \tag{1}$$

## 3.2 FRAMEWORK OF RATIONALE LEARNING

In this paper, we roughly follow the framework of rationale learning presented in Figure 1(b) which consists of the Separator, Rationale & Non-Rationale Mixture and Classifier modules. Differently, in our framework, we additionally consider the partial order relation between the rationale and the non-rationale in the classifier.

### 3.2.1 RATIONALE & NON-RATIONALE SEPARATING

To acquire the rationale and non-rationale subgraphs from the input graph $g_i$ in a batch, we follow (Liu et al., 2022) to use a node-level mask $M_i \in \mathbb{R}^{|\mathcal{V}|}$ indicating the probability of each node in a graph with $|\mathcal{V}|$ nodes belonging to the rationale subgraph:

$$M_i = \sigma(\text{MLP}_1(\text{GNN}_1(g_i))) \in \mathbb{R}^{|\mathcal{V}| \times d}, \tag{2}$$

where $\sigma$ is a sigmoid function, $\text{GNN}_1(\cdot)$ can be any GNN encoders, such as GIN (Xu et al., 2019) or GCN (Kipf & Welling, 2017). Conversely, the probability belonging to the non-rationale graph can be presented as $\mathbf{1}_{|\mathcal{V}|} - M_i$. Then, we use another encoder $\text{GNN}_2$ to obtain the representation $H = \text{GNN}_2(\cdot) \in \mathbb{R}^{|\mathcal{V}| \times d}$ of the node itself. Next, we can get the rationale and non-rationale representation ($r_i$ and $e_i$) in the latent space :

$$r_i = \text{Pooling}(M_i \odot H) \in \mathbb{R}^d, e_i = \text{Pooling}((\mathbf{1}_{|\mathcal{V}|} - M_i) \odot H) \in \mathbb{R}^d, \tag{3}$$

where $\odot$ is the element-wise product to get the nodes representations and $\text{Pooling}$ (e.g., sum pooling) combines them into the graph-level representation.

### 3.2.2 RATIONALE & NON-RATIONALE MIXTURE LEARNING

In this subsection, we illustrate the Rationale & Non-Rationale Mixture method by introducing non-rationale based augmentation methods. The non-rationales can be viewed as spurious correlations or noise of the graphs. In order to enhance the robustness of the model, we combine the anchor rationale $r_i$ with all the other non-rationales $e_j \in E_B = \{e_1, e_2, ..., e_B\}$ in the batch, and we construct the new graph representation $h_i$:

$$h_i = \text{AGG}(r_i, e_j), e_j \in E_B. \tag{4}$$

The combination function $AGG(\cdot)$ can be any combining/pooling function, here we use the element-wise sum pooling. Finally, we can collect a B-size batch of graph mixtures $H = \{h_1, h_2, ..., h_B\}$. Besides, since the rationales do not change, labels of the mixed data remain unchanged. With the mixed data, we can obtain multiple train data and further enhance the robustness of the model.

### 3.2.3 CLASSIFIER AND PARTIAL ORDER LEARNING

After acquiring the rationale and non-rationale representation ($r_i$ and $e_i$), in the classifier, the prediction score $\hat{y}_{r_i}$ and $\hat{y}_{e_i}$ based on them are produced by the classifier $f_C(\cdot)$:

$$\hat{y}_{r_i} = f_C(r_i), \quad \hat{y}_{e_i} = f_C(e_i). \tag{5}$$

Having acquiring the $y_i$ with the rationale $r_i$, the loss function for input graphs $g_i$ in a batch can be defined as:

$$\mathcal{L}_{\mathcal{R}} = - \sum_{r_i \in R_B} (y_i \log \hat{y}_{r_i} + (1 - y_i) \log(1 - \hat{y}_{r_i})). \tag{6}$$

Then, considering the B-size batch of graph mixtures $H = \{h_1, h_2, ..., h_B\}$, we can train the model with the binary cross entropy loss:

$$\mathcal{L}_{\mathcal{M}} = -\frac{1}{B} \sum_{h_i \in H_B} (y_i \log \hat{y}_{h_i} + (1 - y_i) \log(1 - \hat{y}_{h_i})). \tag{7}$$

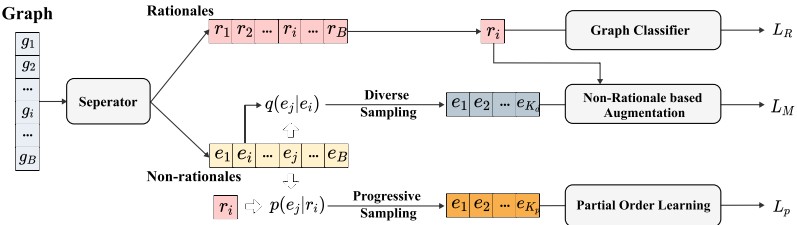

Figure 2: Conditional non-rationale sampling for graph generalization.

Finally, to mine the relative poverty between rationales and non-rationales, in the classifier, we additionally design a Rationale & Non-Rationale partial order learning method. Specifically, in previous works, rationales representations are label-related features while non-rationales representations are label-unrelated features, so we naturally propose to learn the partial order between rationales and non-rationales. This can be defined that the prediction score on the label by rationale $r_i$ should be higher than arbitrary non-rationale $e_j$: $\hat{y}_{r_i} > \hat{y}_{e_j}$.

As for optimization, we use a widely-used pairwise optimization in recommender systems: BPR loss (Rendle et al., 2012; Lian et al., 2020), which maximizes the difference between the predicted probability of a positive pair and a negative pair. Similarly, Rationale & Non-Rationale partial order learning can be formulated in the following to maximize the predicted probability between rationale prediction score $\hat{y}_{r_i}$ and non-rationale prediction score $\hat{y}_{e_j}$:

$$\mathcal{L}_P = -\frac{1}{B}\sum_{i=1}^{B}\ln\sigma(\hat{y}_{r_i} - \hat{y}_{e_j}). \tag{8}$$

### 3.3 CONDITIONAL NON-RATIONALE SAMPLING

From the above framework, we can find in both Rationale & Non-Rationale mixture and partial order learning, how to sample a non-rationale is quite significant. Simple *"random"* sampling may introduce invalid samples, rendering this randomized operation unsuitable. Therefore, in this subsection, we introduce our **Co**mbine and **Co**mpare (**CoCo**) with non-rationales for Graph Rationale Learning method with Conditional Non-rationale Sampling method (Figure 2), which consists of diverse sampling and progressive sampling.

#### 3.3.1 DIVERSE SAMPLING

In section 3.2.2, we have discussed that through replacing the originally separated non-rationale subgraph with the other non-rationale subgraphs can enhance the model's robustness against diverse noise signals. However, the current common *random* sampling methods likely sample the non-rationale $e_j$ which is closed to the non-rationale $e_i$ separated from the input graph $g_i$. It will make very few limited contributions for the classifier to successfully classify the new graph with mixture $h_i$. Therefore, the intention of sampling for graph augmentation is to select diverse non-rationale subgraphs that are far from the anchor separated non-rationales. Here we give a formal definition of *Non-rationales Diversity* of each sample conditional on the original non-rationale $e_i$ below:

**Definition 1** *(Non-rationales Diversity). Given a non-rationales set $E_B = \{e_1, e_2, ..., e_B\}$, the diversity for the non-rationale $e_j$ is defined with the softmax of cosine similarities $\text{sim}(\cdot, \cdot)$ between $e_j$ and $e_i$:*

$$q(e_j|e_i) = 1 - \frac{\text{sim}(e_i, e_j)}{\sum_{e_{j'}\in E_B}\text{sim}(e_i, e_{j'})}, \tag{9}$$

where the larger $q(e_j|e_i)$, the more diverse. Here, there is no need to calculate all non-rationales in a batch. We first randomly select $K_d$ instances into a new non-rationale set $E_{K_d} = \{e_1, e_2, ..., e_{K_d}\}$. Then, the graph mixture set $H_{K_d}$ is generated by Eq.(4) for predicting the label. Finally, by considering this diversity as the confidence weight of each sample, we reformulate the augmentation optimization Eq.(7) in the following way:

$$\mathcal{L}_{\mathcal{M}} = -\sum_{h_i\in H_{K_d}} q(e_j|e_i)\cdot(y_i\log\hat{y}_{h_i} + (1-y_i)\log(1-\hat{y}_{h_i})), \tag{10}$$

which pays more attention to diverse non-rationales. By the way, if the proposal distribution $q(e_j|e_i) = \frac{1}{K_d}$ and $K_d = B$, Eq.(10) is degraded back to Eq.(7).

### 3.3.2 PROGRESSIVE HARD SAMPLING

In this subsection, we introduce the progressive hard sampling concerning Rationale & Non-Rationale partial order learning. The general sample case is randomly sampling the non-rationales separated in a batch to be negative non-rationales to the anchor rationale. However, choosing negatives in the *"random"* manner may not be the best choice. Prior works (Formal et al., 2022; Kalantidis et al., 2020) has shown the effectiveness of hard sample mining in pair-wise learning, which is to find samples that are difficult for the current model to discriminate. Similarly, the hard non-rationales should be close to the current rationale in the latent space. In that, we define non-rationale hardness should be conditional on the current rationale $r_i$ as follows:

**Definition 2** *(Non-rationales Hardness). Given a non-rationales set $E_B = \{e_1, e_2, ..., e_B\}$ and the anchor rationale $r_i$, the hardness for non-rationales $e_j$ is the softmax cosine similarities $\text{sim}(\cdot, \cdot)$ between $r_i$ and arbitrary $e_j$:*

$$p(e_j|r_i) = \frac{\text{sim}(r_i, e_j)}{\sum_{e_{j'} \in E_B} \text{sim}(r_i, e_{j'})}, \tag{11}$$

through the defined hardness, we screen out those non-rationales whose hardness $p(e_j|r_i)$ are too small by setting a hardness lower percentile $p_l \in [0, 100]$, selecting non-rationales whose hardness meets $p(e_j|r_i) > p_l$ .

However, Ridnik et al. (2021); Chen et al. (2021); Ding et al. (2020) point out that sampling hard instances still confronts a big challenge. That is when the representations tend to be unstable in the initial learning, selecting hard 'non-rationales' will exclude rationale in fact. Inspired by Wu et al. (2020a), we set a window percentile range $w_t = [p_l^t, p_u]$ to progressively generate a hard non-rationale set $E_C$:

$$\begin{aligned} w_t &= [p_l^t, p_u], p_l^t, p_u \in [0, 100], \\ E_C &= \{e_1, e_2, ..., e_{K_p}|p(e_j|r_i) \in w_t\}, \end{aligned} \tag{12}$$

where upper percentile $p_u$ is to control the non-rationale to not be too hard concerning the current rational $r_i$. The $p_l^t$ linearly grows up with epoch number $t$ in order to stably learn the partial order between rationales and non-rationales from the easy to hard level.

### 3.4 OPTIMIZATION AND INFERENCE

During the training stage, we train final loss by the Rationales prediction, Non-rationale based Augmentation, and Rationale & Non-rationale partial order learning cooperatively:

$$\mathcal{L}_{\mathcal{F}} = \mathcal{L}_{\mathcal{R}} + \alpha \cdot \mathcal{L}_{\mathcal{M}} + \beta \cdot \mathcal{L}_{\mathcal{P}}, \tag{13}$$

where $\alpha, \beta$ are trade-off hyper-parameters that balances these losses. By the way, the classification loss functions is based on binary classification for illustration. In multi-class classification, we use the Categorican Cross-entropy instead. During the inference stage, the label $y_{r_i}$ predicted by rationale $r_i$ serves as the final predicted label.

## 4 EXPERIMENTS

In this section, we conduct experiments to demonstrate that CoCo can effectively alleviate the OOD problem on graphs. Specifically, we first introduce the experimental setup (Section 4.1), followed by the main results (Section 4.2) and the detailed analyses of our model (Section 4.3 - 4.5).

### 4.1 EXPERIMENTAL SETUP

#### 4.1.1 DATASETS

- **Spurious-Motif** (Ying et al., 2019; Wu et al., 2022b). A synthetic dataset created for the purpose of predicting the category of motifs within each graph. Specifically, each graph consists of two subgraphs: the motif subgraph (represented as Circle, House, Crane, with values R = 0, 1, 2), and the base subgraph (represented as Tree, Ladder, Wheel, with values E = 0, 1, 2). Notably, the motif subgraph is considered as the rationale at the graph label while the base subgraph can be viewed as the non-rationale. To evaluate the effectiveness of CoCo, we manually generate several datasets. Details are in Appendix B.1.

- **MNIST-75SP** (Knyazev et al., 2019). Each graph in this dataset is converted from an image in MNIST (LeCun et al., 1998) using super-pixels. To simulate distribution shifts in the node features, random noises are introduced into the testing set.
- **Open Graph Benchmark (OGBG)** (Wu et al., 2020b). It's a widely used dataset for machine learning on graphs, specifically focusing on molecular property prediction. We used four OGBG-Mol datasets: MolHIV, MolBACE, MolBBBP, and MolSIDER. These datasets are divided using default splits, ensuring that each split contains a distinct set of scaffolds.

More data statistics about the datasets are depicted in Appendix B.2.

### 4.1.2 IMPLEMENTATION DETAILS

We give the detailed configurations of experiments. First, we adopt the same 5-layer GNN (i.e., GIN (Xu et al., 2018)) with hidden dimension 32, 64 and 128 for MNIST-75SP, Spurious-Motif and OGBG, respectively. As for the loss hyper parameter in Eq.(13), we set $\alpha = 0.3$ and $\beta = 1$ for all the datasets. Concerning sampling details, in diverse sampling, we set the $K_b$ as 8 for MNIST-75SP and Spurious-Motif, 16 for OGBG. In progressive hard sampling, the window lower percentile $p_l$ and upper percentile $p_u$ are set as 10 and 90 for all the dataset[1]. As for the maximum training epochs, we set 30 for MNIST-75SP and Spurious-Motif, while 400 for OGBG dataset. We train the model with train set and evaluate on development set after every epoch, and stop training if evaluation value does not increase for a patience epoch number. The patience is set as 10 for MNIST-75SP and Spurious-Motif, while 40 for OGBG dataset. The batch size is set to 256. And the learning rate of the Adam optimizer (Kingma & Ba, 2014) is initialized as 5e-3 for Spurious-Motif, 1e-2 for MNIST-75SP while 1e-3 for OGBG dataset. All the experiments are conducted five times and the performance is reported with the mean and standard deviations results.

### 4.1.3 BASELINES AND METRICS

We compare our model with a wide range of state-of-the-art approaches, as described below:

- **DIR** (Wu et al., 2022b) introduces an invariant learning approach that identifies causal rationales invariant to perturbations by *random* interventions.
- **DisC** (Fan et al., 2022) is a disentangled GNN framework the learns causal and bias subgraphs by synthesizing the counterfactual unbiased training samples.
- **GREA** (Liu et al., 2022) augments original graph by *random* removing and replacing environments to strengthen the rationale representation learning against the noise signal brought by the environment subgraphs.
- **CAL** (Sui et al., 2022) addresses the issue of spurious correlations in Graph Neural Network (GNN) by debiasing the confounding effects shortcut features in the input graph.
- **GSAT** (Miao et al., 2022)learn a invariant subgraphs under distribution shifts by the attention-based inherent Graph Neural Networks (GNNs).
- **DARE** (Yue et al., 2022) is an advanced rationale approach applied in natural language understanding tasks which incorporates disentanglement to enhance the extraction of rationales. Here we extend its application to elucidating GNNs for extensive comparisons.

Based on previous works (Miao et al., 2022), we evaluate models' performance on MNIST-75SP and Spurious-Mot with ACC (accuracy), and on the OGBG-class datasets with ROC-AUC.

### 4.2 MAIN RESULTS

The main results on Spurious-Motif and MINIST-75SP are reported in Table 1, while the results on OGBG are illustrated in Table 2. From these tables, we find that our proposed CoCo method outperforms all baselines in all metrics, except for GSAT on MINIST-75SP, demonstrating the effectiveness of our proposed conditional non-rationale sampling method.

More specifically, on the Spurious-Motif dataset, CoCo demonstrates a consistently superior performance. Especially when the bias is $0.5$, CoCo exhibits a remarkable improvement of $12.85\%$ in accuracy compared to the state-of-the-art baseline (i.e., DARE). When considering the performance MNIST-75sp dataset, GAST surpasses all other methods by a large margin. This superior performance may be attributed to the unique compatibility between GAST and the MNIST-75SP dataset,

---

[1]As we introduced in Section 3.3.2, the lower percentile $p_l$ linearly grows, i.e., $p_l^t = p_l + (\frac{t}{T}) \cdot (p_u - p_l)$, where t is the current epoch number, T is the maximum training epochs.

Table 1: The graph classification accuracy (mean±std%, the best results are bolded) on MNIST-75SP and Spurious-Motif.

| Method | Spurious-Motif | | | MNIST-75SP |
|--------|-------------|------------|------------|------------|
| | bias = 0.5 | bias = 0.7 | bias = 0.9 | |
| GIN | $0.4444 \pm 0.0621$ | $0.4891 \pm 0.0761$ | $0.4131 \pm 0.0652$ | $0.1201 \pm 0.0042$ |
| DisC | $0.4585 \pm 0.0660$ | $0.4885 \pm 0.1154$ | $0.3859 \pm 0.0400$ | $0.1262 \pm 0.0113$ |
| DIR | $0.3950 \pm 0.0471$ | $0.3872 \pm 0.0531$ | $0.3768 \pm 0.0447$ | $0.1893 \pm 0.0458$ |
| GREA | $0.4251 \pm 0.0458$ | $0.5331 \pm 0.1509$ | $0.4568 \pm 0.0779$ | $0.1172 \pm 0.0021$ |
| CAL | $0.4734 \pm 0.0681$ | $0.5541 \pm 0.0323$ | $0.4474 \pm 0.0128$ | $0.1258 \pm 0.0123$ |
| GSAT | $0.4517 \pm 0.0422$ | $0.5567 \pm 0.0458$ | $0.4732 \pm 0.0367$ | $\mathbf{0.2381 \pm 0.0186}$ |
| DARE | $0.4843 \pm 0.1080$ | $0.4002 \pm 0.0404$ | $0.4331 \pm 0.0631$ | $0.1201 \pm 0.0042$ |
| **CoCo(Ours)** | $\mathbf{0.6128 \pm 0.0585}$ | $\mathbf{0.5964 \pm 0.0449}$ | $\mathbf{0.4896 \pm 0.0440}$ | $0.1946 \pm 0.0249$ |

Table 2: The graph classification AUC (mean±std%, the best results are bolded) on OGBG datasets.

| Method | MolHIV | MolBBBP | MolBACE | MolSIDER |
|--------|--------|---------|---------|----------|
| GIN | $0.7447 \pm 0.0293$ | $0.6584 \pm 0.0224$ | $0.8047 \pm 0.0172$ | $0.5977 \pm 0.0176$ |
| DisC | $0.7731 \pm 0.0101$ | $0.6963 \pm 0.0206$ | $\mathbf{0.8293 \pm 0.0171}$ | $0.5846 \pm 0.0169$ |
| GREA | $0.7714 \pm 0.0153$ | $0.6953 \pm 0.0229$ | $0.8187 \pm 0.0195$ | $0.5864 \pm 0.0052$ |
| DIR | $0.6303 \pm 0.0607$ | $0.6460 \pm 0.0139$ | $0.7391 \pm 0.0282$ | $0.4989 \pm 0.0115$ |
| CAL | $0.7339 \pm 0.0077$ | $0.6582 \pm 0.0397$ | $0.7848 \pm 0.0107$ | $0.5965 \pm 0.0116$ |
| GSAT | $0.7524 \pm 0.0166$ | $0.6722 \pm 0.0197$ | $0.7021 \pm 0.0354$ | $0.6041 \pm 0.0096$ |
| DARE | $0.7836 \pm 0.0015$ | $0.6820 \pm 0.0246$ | $0.8239 \pm 0.0192$ | $0.5921 \pm 0.0260$ |
| **CoCo(Ours)** | $\mathbf{0.8053 \pm 0.0135}$ | $\mathbf{0.7077 \pm 0.0073}$ | $0.8275 \pm 0.0129$ | $\mathbf{0.6052 \pm 0.0160}$ |

as its performance appears to degrade significantly when applied to other datasets. In the case of the real-world datesets OGBG, CoCo also achieves really advanced performance, on the MolHIV, MolBBBP and MolSider sub-datasets. These findings sufficiently prove that our proposed CoCo method can alleviate the distribution shifts between the train set and test set.

### 4.3 COMPONENT EFFECTIVENESS

To validate the effectiveness of each component we designed in CoCo, we conduct experiments on the real-word dataset OGBG with several ablated variants. Specifically, We disassemble CoCo by removing the rationale & non-rationale mixture learning (Eq.(10), CoCo-M), and Partial order learning (Eq.(8), CoCo-P). As illustarted in Table 3, The performance of two variants, CoCo-P and CoCo-M, shows marked decreases, demonstrating the vital role each module plays in the overall system. The worst performance is observed in CoCo-PM, which removes both two modules, further validating the necessity and non-redundancy of our design. Interestingly, the performance declines of CoCo-P and CoCo-M are close, suggesting these two sampling strategy (i.e., diversive sampling and progressive sampling) play equally significant roles in the overall performance.

### 4.4 MODEL SENSITIVITY

**Diverse Sampling Sensitivity.** As we introduced in Section 3.3.1, we sample $K_d$ non-rationales in a batch for graph augmentation. In this part, we investigate the sensitivity of sample number $K_d$ of CoCo. Figure 3 (a) shows the CoCo's performance on Spurious-Motif (Bias=0.7) and OGBG-MolBBBP's results are in Appendix C.2. The ACC/AUC performance initially increases, peaks, and then either decreases or stabilizes. This trend can be explained by our selection of $K_d$ non-rationale samples in a batch. As $K_d$ increases beyond a certain point, it introduces some samples that are not non-rationales. This inclusion can disrupt the non-rationale based augmentation process, leading to a potential decrease or plateau in performance. Besides, we show the performance of CoCo-random by replacing the diversity sampling with random sampling. We find that CoCo performs well when the sample number is small, while CoCo-random requires more non-rationale instances to enrich graph augmentation. It verifies that the non-rationale based augmentation equipped with diverse sampling significantly aids the learning process. In this way, CoCo can be applied to more situations, especially when the batch size cannot scale accordingly with the dataset scales.

**Progressive Hard Sampling Sensitivity.** In progressive hard sampling part (Section 3.3.2), we set a window percentile range $[p_l^t, p_u]$ to progressively generate a hard non-rationale set, where upper percentile $p_u$ is to control the non-rationale to not be too hard. In Table 4, we change the hyperparameter $p_u$ and report the performance on the Spurious-Motif

Table 4: The mean AUC of window upper-percentile on Spurious-Motif dataset.

| Datasets | Bias=0.5 | Bias=0.7 | Bias=0.9 |
|----------|----------|----------|----------|
| CoCo ($p_u$=70) | 0.5694 | 0.5556 | 0.4748 |
| CoCo ($p_u$=80) | 0.5717 | 0.5484 | 0.4796 |
| CoCo ($p_u$=100) | 0.5937 | 0.5285 | 0.4503 |
| CoCo ($p_u$=90) | **0.6128** | **0.5964** | **0.4896** |

dataset. From this table, we find that too larger or smaller $p_u$ will both lead to obvious decreases in model performance. This observation aligns with theoretical expectations. On the one hand,

Table 3: The mean accuracy performance of ablated variants on OGBG dataset.

| Datasets | MolHiv | MolBBBP | MolBACE | MolSIDER |
|----------|--------|---------|---------|----------|
| CoCo-PM | $0.7624 \pm 0.0116$ | $0.6560 \pm 0.0157$ | $0.7892 \pm 0.0031$ | $0.5776 \pm 0.0209$ |
| CoCo-P | $0.7899 \pm 0.0146$ | $0.6792 \pm 0.0132$ | $0.8068 \pm 0.0184$ | $0.5825 \pm 0.0224$ |
| CoCo-M | $0.7845 \pm 0.0254$ | $0.6673 \pm 0.0067$ | $0.7964 \pm 0.0208$ | $0.5837 \pm 0.0114$ |
| CoCo | $\mathbf{0.8053 \pm 0.0135}$ | $\mathbf{0.7077 \pm 0.0073}$ | $\mathbf{0.8275 \pm 0.0129}$ | $\mathbf{0.6052 \pm 0.0160}$ |

**(a) Effects on Spurious-Motif**  **(b) Loss Hyperparameters**  **(c) Rationale Performance**  **(d) Motif: House Base: Tree&Wheel**

Figure 3: (a) Effects of the non-rationale number in diverse sampling; (b) Effects of parameter $\alpha$ and $\beta$ related to loss $\mathcal{L}_{\mathcal{M}}$ and $\mathcal{L}_{\mathcal{P}}$; (c)Rationale Performance on OGBG-MolBBBP; (c) Visualization of Spurious-Motif testing dataset.

smaller $p_u$ excludes these challenging non-rationale examples, thereby limiting the model's capacity to reach its full potential. On the other hand, a significantly larger $p_u$ may introduce excessively difficult samples prematurely, which may be rationales in reality according to hard sampling researches (Robinson et al., 2020; Chen et al., 2021) and disrupt the model's training process.

**Loss Hyper-parameter Sensitivity.** We further analyze the CoCo's performance along with the hyperparameter $\alpha$ and $\beta$ in the objective function ( Eq.(13)). Specifically, we fix one hyper parameter while investigating the effect of the other. The results are illustrated in Figure 3(b). We observe both $\alpha$ and $\beta$ significantly affect the final performance, as they modulate the weights of the two losses, thus directly impacting the learning process. In the experiments, we tune them on the development set to obtain the best settings ($\alpha$=0.3, $\beta$=1.0), which is consistent with the results in Figure 3.

### 4.5 RATIONALE ANALYSIS

**Rationale Performance.** To demonstrate the CoCo's ability to separate rationales from non-rationales, we draw the accuracy score using the rationale features ($y_{r_i}$ in Section 3.2.3) with the increasing of the epoch number on OGBG dataset. As shown in Figure 3(c) , we further make comparisons with GREA (Liu et al., 2022) used the similar paradigm. This method designs random sampling strategy to empower the rationale learning process. From this figure, we find that, compared with GREA, CoCo not only achieves higher AUC earlier (except a short period of lag), but also ultimately achieves a better performance. It is reasonable as, in the initial stage, these two sampling methods (diverse sampling and progressive hard sampling) haven't worked well due to the not good enough representations. As the training progresses, two sampling methods come to play roles in rationale learning, and finally contribute to the better performance compared to GREA. This demonstrates CoCo's excellent ability of separating rationales from the input graphs.

**Visualization.** To better illustrate the effectiveness of rationale learning, we show three visualization cases from Spurious-Motif testing datasets in Figure 3(d) and Figure 5 in Appendix C.2. Specifically, the nodes and edges highlighted by green colors belong to the recognized rationale subgraphs. We can observe that CoCo exhibits excellent ability for identifying the rationale subgraphs even combined with diverse non-rationales. For instance, the Motif House subgraph is standout in both the Base Tree and Wheel (Figure 3(d)). This proves that our method can enhance the separator's ability to extract the rationale, even in the face of unseen disturbances.

### 5 CONCLUSION

In this paper, to solve the graph out-of-distribution (OOD) problem, we proposed a Combine and Compare (CoCo) with non-rationales for graph rationale learning method with conditional non-rationale sampling. Specifically, CoCo first employed a separator to decompose the input graph into rationale and non-rationale subgraphs. Then, we introduced a diverse sampling method to sample non-rationales and combined them with the rationale to achieve non-rationale based augmentation. Further, multiple training data could be obtained. Finally, CoCo yielded the prediction results based on both rationales and combined data. Meanwhile, considering the partial order relationship of rationales and non-rationales, we developed a progressive hard sampling to detect the negative non-rationales to the anchor rationale, enhancing the model's ability to discriminate them. Extensive experiments on both benchmarks and synthetic datasets validated the effectiveness of CoCo.

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

# A PSEUDO CODE OF CONDITIONAL NON-RATIONALE SAMPLING METHOD

Here we show the pseudo code of conditional non-rationale sampling method in the training stage.

---

**Algorithm 1** Conditional Non-Rationale Sampling in Graph Rationale Learning

---

**Input**: Training set of labeled graphs $\mathcal{G} = \{(g_i, y_i)\}_i^n$
**Output**: Prediction $\hat{y}_r$ with graph rationalization
Initialize parameters of separator $f_S(\cdot)$ and classifier $f_C(\cdot)$ randomly;

1: **while** not converge **do**
2:     Sample a batch of graph $G_B \in G$ of size B.
3:     Separate the input graph $G_B = \{g_i\}_i^B$ into the rationale $R_B = \{r_i\}_i^B$ and environment $E_B = \{e_i\}_i^B$ in the latent space through separator $f_S(\cdot)$. (Eq.(2) and (3))
4:     **for** each rationale representation $r_i \in R_B$ **do**
5:         Get the rationale prediction $\hat{y}_{r_i}$ and Calculate the loss $\mathcal{L}_\mathcal{R}$ with Eq.(6)
        # Non-rationale based augmentation with diverse sampling
6:         Get non-rationale set $E_{K_d} = \{e_j\}_j^{K_d}$ for augmentation
7:         Construct a new graph $h_i$ by combining the anchor rationale $r_i$ and non-rationale $e_j$
8:         Calculate the diversity of $e_j$ conditional in the complement environment $e_i$ of the anchor rationale $r_i$ based on Eq.(9)
9:         Get the diversity weighted loss $\mathcal{L}_\mathcal{M}$ with Eq.(10)
        # Rationale & non-rationale partial order learning with progressive hard sampling
10:       Calculate the hardness $p(e_j|r_i)$ of non-rationale $e_j$ conditional on the anchor rationale $r_i$ based on Eq.(11)
11:       Set a window percentile range $w_t = [p_l^t, p_u]$ to generate the non-rationale set $E_C$ for partial order learning based on Eq.12
12:       Get the partial order loss $\mathcal{L}_\mathcal{P}$ with Eq.(8)
13:       Update parameters w.r.t. the final loss Eq.(13)
14:     **end for**
15: **end while**

---

# B DATA DESCRIPTION

## B.1 DATA SYNTHESIZATION OF SPURIOUS-MOTIF

When creating the Spurious-Motif dataset, we initially generate the training dataset by uniformly sampling each motif while controlling the distribution of the base. This distribution, denoted as P(E), is determined by the bias parameter $b$, and it is defined as follows: $P(E) = b \times I(E = R) + (1 - b)/2 \times I(E \neq R)$, where $b$ regulates the degree of spurious correlation. In this study, we used $b$ values of 0.5, 0.7, and 0.9 in the training dataset. For the test dataset, we randomly pair motifs and bases, with $b$ set to 1/3.

## B.2 DATA STATISTICS

The following tables are the statistics of experimental datasets we use for validating the effectiveness of CoCo, including one synthetic datasets (i.e., Spurious-Motif) and two real-word datasets (i.e., MNIST-75SP and OGBG).

Table 5: The statistics of Spurious-Motif and MNIST-75SP datasets.

| Datasets | Spurious-Motif bias = 0.5 | bias = 0.7 | bias = 0.9 | MNIST-75SP |
|---|---|---|---|---|
| #Graphs(Train/Val/Test) | 3,000/3,000/6,000 | 3,000/3,000/6,000 | 3,000/3,000/6,000 | 5,000/1,000/1,000 |
| Avg #nodes | 29.6 | 30.8 | 29.4 | 66.8 |
| Avg #edges | 42.0 | 45.9 | 42.5 | 600.2 |
| Classes | 3 | 3 | 3 | 10 |

Table 6: The statistics of OGBG datasets.

| Datasets | OGBG | | | |
|---|---|---|---|---|
| | MolHIV | MolBBBP | MolBACE | MolSIDER |
| #Graphs(Train/Val/Test) | 32,901/4,113/4,113 | 1,631/204/204 | 1,210/151/152 | 1,141/143/143 |
| Avg #nodes | 25.5 | 34.12 | 24.1 | 33.6 |
| Avg #edges | 27.5 | 26.0 | 36.9 | 35.4 |
| Classes | 2 | 2 | 2 | 27 |

# C  MORE EXPERIMENTAL RESULTS

## C.1  RESULTS ON GCN BACKBONE

As introduced in Section 4.1, we adopt the GIN (Xu et al., 2018) as the backbone of our CoCo model. In this section, we replace GIN with GCN (Kipf & Welling, 2017) and report the main results in Table 7 and 8. From these results, The GCN-based CoCo also achieves the optimal performance in most cases, further underscoring the superiority of our design. This observation aligns with the analyses presented in Section 4.2, which substantiates the robust generalization capability of the proposed CoCo model architecture.

Table 7: The graph classification accuracy (mean±std%, the best results are bolded) on testing sets of MNIST-75SP and Spurious-Motif.

| Method | MNIST-75SP | Spurious-Motif | | |
|---|---|---|---|---|
| | | bias = 0.5 | bias = 0.7 | bias = 0.9 |
| GCN | $0.1201 \pm 0.0042$ | $0.4281 \pm 0.0520$ | $0.4471 \pm 0.0312$ | $0.4588 \pm 0.0840$ |
| DisC | $0.1262 \pm 0.0113$ | $0.4698 \pm 0.0408$ | $0.4312 \pm 0.0358$ | $0.4713 \pm 0.1390$ |
| GREA | $0.1172 \pm 0.0021$ | $0.4687 \pm 0.0855$ | $0.5467 \pm 0.0742$ | $0.4651 \pm 0.0881$ |
| DIR | $0.1283 \pm 0.1283$ | $0.4281 \pm 0.0520$ | $0.4471 \pm 0.0312$ | $0.4588 \pm 0.0840$ |
| CAL | $0.1258 \pm 0.0123$ | $0.4091 \pm 0.0398$ | $0.3772 \pm 0.0763$ | $0.3566 \pm 0.0323$ |
| GSAT | $\mathbf{0.2381 \pm 0.0186}$ | $0.3630 \pm 0.0444$ | $0.3601 \pm 0.0419$ | $0.3929 \pm 0.0289$ |
| DARE | $0.1231 \pm 0.0062$ | $0.4609 \pm 0.0648$ | $0.5035 \pm 0.0247$ | $0.4494 \pm 0.0526$ |
| **CoCo-GCN(Ours)** | $0.2031 \pm 0.0642$ | $\mathbf{0.5764 \pm 0.0989}$ | $\mathbf{0.5804 \pm 0.0792}$ | $\mathbf{0.4993 \pm 0.1154}$ |

Table 8: The graph classification AUC (mean±std%, the best results are bolded) on testing sets of OGBG datasets.

| Method | OGBG | | | |
|---|---|---|---|---|
| | MolHIV | MolBBBP | MolBACE | MolSIDER |
| GCN | $0.7128 \pm 0.0188$ | $0.6665 \pm 0.0242$ | $0.8135 \pm 0.0256$ | $0.6108 \pm 0.0075$ |
| DisC | $0.7791 \pm 0.0137$ | $0.7061 \pm 0.0105$ | $0.8104 \pm 0.0202$ | $0.6110 \pm 0.0091$ |
| GREA | $0.7816 \pm 0.0079$ | $0.6970 \pm 0.0089$ | $0.8044 \pm 0.0063$ | $0.6133 \pm 0.0239$ |
| DIR | $0.4258 \pm 0.1084$ | $0.5069 \pm 0.1099$ | $0.7002 \pm 0.0634$ | $0.5224 \pm 0.0243$ |
| CAL | $0.7501 \pm 0.0094$ | $0.6635 \pm 0.0257$ | $0.7802 \pm 0.0207$ | $0.5559 \pm 0.0151$ |
| GSAT | $0.7598 \pm 0.0085$ | $0.6437 \pm 0.0082$ | $0.7141 \pm 0.0233$ | $0.6179 \pm 0.0041$ |
| DARE | $0.7523 \pm 0.0041$ | $0.6823 \pm 0.0068$ | $0.8066 \pm 0.0178$ | $0.6192 \pm 0.0079$ |
| **CoCo-GCN(Ours)** | $\mathbf{0.7992 \pm 0.0084}$ | $\mathbf{0.7075 \pm 0.0081}$ | $\mathbf{0.8167 \pm 0.0233}$ | $\mathbf{0.6199 \pm 0.0063}$ |

## C.2  EFFECTS OF SAMPLE NUMBER ON OGBG-MOLBBBP

Figure 4 shows that the real-world dataset OGBG-MolBBBP achieves high and stable performance when the sample number $K_d$ increased.

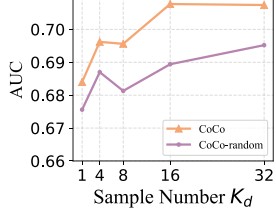

Figure 4: The effects of sample number on OGBG-MolBBBP

## C.3    MORE VISUALIZATIONS ON SPURIOUS-MOTIF TESTING DATASET

Besides examples in Figure 3, we demonstrate more visualizations cases here. Both the Motif Circle and Crane rationale subgraphs are extracted out from Base Tree and Wheel successfully, further validating separator's CoCo strong ability to extract the rationales.

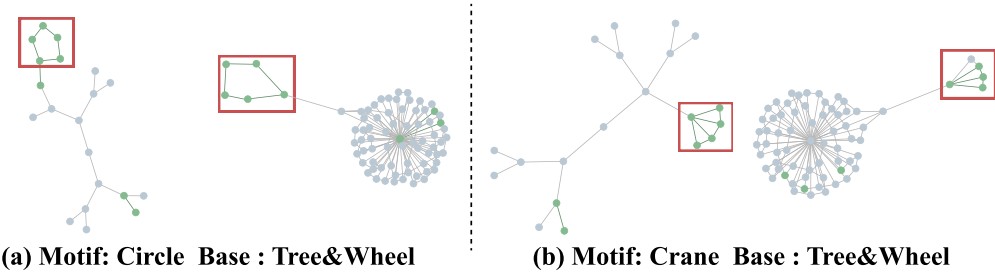

**(a) Motif: Circle  Base : Tree&Wheel**                **(b) Motif: Crane  Base : Tree&Wheel**

Figure 5: Visualizations of CoCo rationales in Spurious-Motif testing dataset, where the recognized rationales are highlighted by green colors.

