# OpenReview forum: "Combine and Compare: Graph Rationale Learning with Conditional Non-Rationale Sampling"
_ICLR.cc/2024/Conference — ICLR 2024 Conference Withdrawn Submission_

### Official Review · Reviewer_U9wj · 2023-10-30

**Soundness:** 3 good
**Presentation:** 3 good
**Contribution:** 2 fair
**Rating:** 5
**Confidence:** 4

**Summary:**

In this paper, the authors propose a new method named Combine and Compare (CoCo) for graph out-of-distribution (OOD) generalization. The proposed method is based on rational-based methods. Specifically, it proposes a diverse sampling to avoid duplicate non-rationales. Besides, it proposes a non-rationale progressive hard sampling to decorrelate hard non-rationales. Experimental results demonstrate the effectiveness of the proposed method.

**Strengths:**

(1)	Graph OOD generalization is an important and trending topic, the rational-based methods are promising for tackling graph OOD generalization.
(2)	The proposed method is intuitive to understand, and the paper is well-written in general.
(3)	The authors have provided the source codes for reproducibility.

**Weaknesses:**

(1)	Though I acknowledge the proposed method is a valid solution, the technical contribution of the paper is limited as the proposed method is largely based on heuristics, i.e., the diverse sampling and progressive hard sampling. It would make the paper stronger if more theoretical analyses could be provided to demonstrate the effectiveness of the proposed method in theory, e.g., the diverse sampling and progressive hard sampling could facilitate in identifying rationales or improving OOD generalization (maybe under some assumptions).
(2)	Though somewhat loosely connected, I feel the progressive hard sampling seems to be related to the curriculum learning literature, where a hardness metric is also proposed and the samples are optimized based on the difficulty. The authors may want to discuss the differences with these methods, which are also studied in graph machine learning.
(3)	Since the overall goal of the progressive hard sampling is based on “non-rationales and labels should be de-correlated compared to the rationales”, I wonder whether the authors have considered directly using de-correlation methods such as sample re-weighting, which are widely studied in the recent causal-based machine learning literature.
(4)	There is no discussion regarding the complexity/efficiency of the proposed method, which could be added.
(5)	The authors focus on graph-level tasks such as graph classification. I wonder whether the proposed method can be applied to other tasks, such as link prediction and node classification.

**Questions:**

See Weaknesses above

---

### Official Review · Reviewer_gNbk · 2023-10-30

**Soundness:** 2 fair
**Presentation:** 2 fair
**Contribution:** 2 fair
**Rating:** 3
**Confidence:** 4

**Summary:**

The performance of Graph Neural Networks (GNNs) drops drastically when the distribution shift occurs between the training and testing datasets. Recent work employs the graph rationale learning approach to tackle this issue. However, this approach faces 1). duplicate samples in augmentation and 2). not considering the de-correlation between labels and non-rationales. This work proposes a non-rationale sampling scheme to improve graph rationale learning. The whole framework is built upon [1] to divide the input graphs into rationales and non-rationales. Then, it proposes progressive sampling and diverse sampling strategies to further improve the performance of the proposed method.

[1]. Graph rationalization with environment-based augmentations. KDD 2022.

**Strengths:**

Improving the out-of-distribution (OOD) generalization of GNNs is crucial to implementing GNNs in high-stake scenarios. This work will make more people familiar with this topic.

**Weaknesses:**

However, there are several concerns regarding the literature review, novelty of methodology, and empirical evaluation.

Literature Review:

1. A discussion on related works in graph rationale learning is lacking. For example, recent works can be categorized into information-theoretic approaches (GIB [1], GSAT [2]), invariant learning (DIR [3]), and mixture approaches (CIGA [4]). These approaches, although not originally designed for OOD generalization, have been proven to be robust to distribution shifts. Thus, it is necessary to discuss the connection between the proposed method and these works in graph rationale learning.

2. A literature review on diverse sampling on graphs is necessary since this approach is widely employed in graph self-supervised learning [5] and graph augmentation [6,7].

Methodology:

1. This work aims to address two limitations of the previous graph rationale learning method: 1). duplicate samples in augmentation and 2). not considering the de-correlation between labels and non-rationales. Although 1) can be solved by diverse sampling under some mild assumptions, 2) seems to still remain unsolved. Is there any justification for how the proposed method solves 2)?

2. Regarding limitation 2) above, previous information-theoretic approaches [1,2] have already provided theoretical justification that such de-correlation is achieved by optimizing an information-bottleneck objective. Is there any possibility of integrating the proposed method with the information-theoretic approaches to address limitation 2)?

3. The reviewer is also concerned with using Eqn. 4 to obtain the embeddings of the augmented graphs, since the node embedding should be re-computed before aggregating for the graph embeddings.

Experiments:

1. The reviewer is curious why DIR underperforms GIN model in Table 1. Since DIR is designed for graph OOD generalization but GIN is a general GNN architecture, it is supposed that DIR should outperform GIN on most datasets.

Reference (Minor): Duplicate reference on "How powerful are graph neural networks"




[1]. Graph Information Bottleneck for Subgraph Recognition. ICLR 2021.
[2]. Interpretable and generalizable graph learning via stochastic attention mechanism. ICML 2022.
[3]. Discovering invariant rationales for graph neural networks. ICLR 2022.
[4]. Learning Causally Invariant Representations for Out-of-Distribution Generalization on Graphs. NeurIPS 2022.
[5]. Graph Contrastive Learning with Augmentations. NeurIPS 2020.
[6]. Mind the label shift of augmentation-based graph OOD generalization. CVPR 2023.
[7]. Adversarial causal augmentation for Graph Covariate Shift. Arxiv 2022.

**Questions:**

The authors are encouraged to address the concerns in Weaknesses.

---

### Official Review · Reviewer_S5nv · 2023-11-01

**Soundness:** 2 fair
**Presentation:** 2 fair
**Contribution:** 2 fair
**Rating:** 3
**Confidence:** 4

**Summary:**

This paper proposes a combine and compare method called CoCo with non-rationales for graph rationale learning method with the conditional non-rationale sampling. The method first employs the diverse sampling method to sample non-rationales, avoiding sampling duplicate non-rationales. It introduces a non-rationale progressive hard sampling method to de-correlate hard non-rationales and labels, enhancing the model’s discrimination ability. The method is validated on some benchmark datasets and shows better results.

**Strengths:**

1. This paper researches a very interesting and important problem in the community, which is the OOD problem on graphs.
2. The proposed method is easy to understand for its framework and formal definition of some concepts.
3. The experiments show improvements against the baseline methods, validating the effectiveness of the method.

**Weaknesses:**

1. The novelty of this paper is limited. For example, the proposal in section 3,2 is similar to the existing work DIR [1]. The main contributions in section 3.3 have weak connections with the graph itself but for general machine learning.
2. The motivations on why the proposed can handle OOD graph are not clear. The time complexity and practical efficiency of the method are not present in detail.
3. The experiments are not convincing enough. The considered datasets are limited. I strongly suggest the authors consider more datasets from GOOD (A Graph Out-of-Distribution Benchmark, NeurIPS 2022). The compared baselines are not state-of-the-art, and more baselines should be compared (such as CIGA [2], G-Mixup [3], etc.) The improvements on some datasets are not very significant.

References:

[1] Discovering Invariant Rationales for Graph Neural Networks. ICLR 2022.

[2] Learning Causally Invariant Representations for Out-of-Distribution Generalization on Graphs. NeurIPS 2022.

[3] G-Mixup: Graph Data Augmentation for Graph Classification. ICML 2022.

**Questions:**

Please refer to the main points in weaknesses part

---

### Official Review · Reviewer_2agF · 2023-11-07

**Soundness:** 2 fair
**Presentation:** 1 poor
**Contribution:** 1 poor
**Rating:** 3
**Confidence:** 4

**Summary:**

This paper argues that existing graph rationale learning methods suffer from the following two issues: 1) produce duplicate samples; 2) the relationship between the rationales, non-rationales, and labels is not properly considered. To solve these issues, they propose a new framework, CoCo, including Separator, Rationale & Non-Rationale Mixture and Classifier modules. CoCo employs the diverse sampling method to sample non-rationales and non-rationale progressive hard sampling, avoiding sampling duplicate non-rationales and de-correlating hard non-rationales and labels.

**Strengths:**

1. A large number of experiments and detailed analysis were carried out in the paper.
2. The code of the paper is released, and reproducibility is guaranteed.

**Weaknesses:**

1. The main motivation of this paper is questionable. I list the following reasons:
- The authors argue that existing methods [1-5] usually use random combination strategies to produce "duplicate samples". Why do these duplicate samples affect rational learning or generalization? The author should give some more detailed explanations, such as statistical failure cases or theoretical discussions.
- The authors argue that the existing methods do not de-correlate non-rationales from labels. However, these methods are mainly based on the ideas of invariant learning or causal intervention, aiming to disconnect the backdoor path between environmental features and labels so that the spurious correlation between the environmental features and the label is released. Why do these existing methods fail? The authors do not provide any theoretical or experimental explanations.

2. The novelty of this work is limited, and the technique is incremental. I list the following reasons:
- The proposed framework uses Separator, Rationale & Non-Rationale Mixture and Classifier modules as the backbone. However, these components and modules have been frequently used in a large number of studies [1-5].
- Separator follows a very similar approach to existing work, such as [1-5].
- Rationale & Non-Rationale Mixture adopts a similar approach to studies [1,3,4].
- As can be seen from Figure 2, this work seems to be very similar to the framework figure in GREA [3].

3. This paper is not well presented and contains many writing errors. For example:
- "…are independent and identically distributed distribution…". What is "distributed distribution"?
- “To validate the effectiveness of each component we designed in CoCo, we conduct experiments on the real-word dataset OGBG with several ablated variants.”
- There is no distinction between scalars, vectors, matrices or tensors in mathematical formulas, making it difficult for readers to understand.

[1] Debiasing Graph Neural Networks via Learning Disentangled Causal Substructure, NeurIPS 2022
[2] Learning Invariant Graph Representations for Out-of-Distribution Generalization, NeurIPS 2022
[3] Graph Rationalization with Environment-based Augmentations, KDD 2022
[4] Causal Attention for Interpretable and Generalizable Graph Classification, KDD 2022
[5] Discovering Invariant Rationales for Graph Neural Networks, ICLR 2022
[6] Let Invariant Rationale Discovery Inspire Graph Contrastive Learning, ICML 2022

**Questions:**

Please refer to weaknesses.

**Details Of Ethics Concerns:**

No.